

# Constraints on physical computers in holographic spacetimes

Aleksander M. Kubicki[1], Alex May[2,3]⋆ and David Pérez-Garcia[1,4]†

**1** Department of Applied Mathematics and Mathematical Analysis,
Universidad Complutense de Madrid, 28040 Madrid, Spain
**2** Stanford Institute for Theoretical Physics, Stanford University,
382 Via Pueblo Mall, Stanford, CA 94305-4060, U.S.A.
**3** Perimeter Institute for Theoretical Physics, Waterloo, Ontario N2L 2Y5, Canada
**4** Instituto de Ciencias Matemáticas, 28049 Madrid, Spain

⋆ amay@perimeterinstitute.ca , † dperezga@ucm.es

## Abstract

Within the setting of the AdS/CFT correspondence, we ask about the power of computers in the presence of gravity. We show that there are computations on $n$ qubits which cannot be implemented inside of black holes with entropy less than $O(2^n)$. To establish our claim, we argue computations happening inside the black hole must be implementable in a programmable quantum processor, so long as the inputs and description of the unitary to be run are not too large. We then prove a bound on quantum processors which shows many unitaries cannot be implemented inside the black hole, and further show some of these have short descriptions and act on small systems. These unitaries with short descriptions must be computationally forbidden from happening inside the black hole.

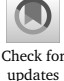

# 1 Introduction

Complexity theory deals with the power of mathematical models of computation. It is generally believed that these models capture the computational abilities of physical computers, but making this connection precise is difficult. For instance, considering a quantum circuit model we may be tempted to equate circuit depth with the time needed to implement the computation on a physical computer. By assuming a bound on energy, that connection can be made precise via the Margolus-Levinin theorem [1]. For any given unitary however, a Hamiltonian can be constructed which implements that unitary arbitrarily quickly, even at bounded energy [2]. This means that in this Hamiltonian model of computation, an energy bound doesn't suffice to relate computational and physical notions of time. Observations such as this one leave it unclear how to connect the limits of physical computers and mathematical models of computation.

In this article we make a preliminary step towards understanding the limits of physical computers. To consider the full set of constraints on physical computers, and the full physical setting that can be exploited by a computer, we consider computation in the context of quantum gravity. We work within the framework of AdS/CFT, which claims an equivalence between quantum gravity in asymptotically anti de Sitter (AdS) spaces and a purely quantum mechanical theory (a conformal field theory, the CFT) living at the boundary of that spacetime. Our main result is a construction of a family of unitaries that a computer operating inside of a black hole with entropy $S_{bh}$ cannot perform, where the computation is on $n$ qubits with $\log S_{bh} \leq n \ll S_{bh}$ and the family we construct is of size $2^{o(S_{bh})}$. Because $n \ll S_{bh}$, the inputs to the computation do not themselves couple strongly to gravity. Instead, it must be the computation on these small inputs that is restricted.

While we are ultimately interested in the physical limits of computers in our universe, working within the context of the AdS/CFT correspondence gives us a precise framework for quantum gravity. As well, a fundamental observation in computer science is that the power of computers is robust to "reasonable" changes in the details of the computing model: classical computers can be described in terms of Turing machines, uniform circuits, etc. and the resources needed to solve a given computational problem will change only polynomially. Quantum computers are similarly robust. This robustness suggests understanding the power of computers in AdS is likely to yield insights that apply more broadly.

Naively, the AdS/CFT duality between a bulk quantum gravity theory and quantum mechanical boundary suggests the power of computers in quantum gravity should be equivalent in some way to quantum computers. We can imagine simulating the CFT on a quantum computer, and thereby producing the outcomes of any computations run in the dual bulk picture. This approach is complicated however by the possibility that mapping between the boundary CFT descriptions and bulk gravity description is exponentially complex [3–6]. Consequently determining the result of the bulk computation from the boundary simulation may itself be highly complex, allowing for a discrepancy in efficiencies between the bulk and boundary. An intriguing observation is that this leaves open the possibility of a quantum gravity computer being much more powerful than a quantum computer [7].

In this work, we give a strategy to restrict bulk computation using the existence of the boundary quantum mechanical description. The crucial property of the bulk to boundary map we assume is state independence, which we have in AdS/CFT when reconstructing suitably small bulk subsystems. We also use that this map is isometric.[1] The state independence of

---

[1]For experts, one comment is that in our context we are assured that the relevant bulk states are physical states in the fundamental description. We are *not* claiming that the entire bulk effective field theory (EFT) Hilbert space can be recovered from the CFT. In fact, the map from the EFT Hilbert space to the CFT Hilbert space is expected to be non-isometric [8].

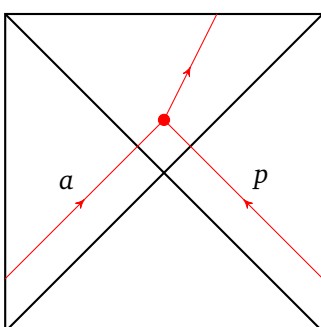

Figure 1: A two sided black hole, with systems $a$ and $p$ falling in from opposite sides. The state on $P$ describes a unitary, which should be applied to the state on $A$.

the bulk to boundary map allows us to relate bulk computation to programmable quantum processors, a well studied notion in quantum information theory. Using tools from functional analysis, we give a bound on the average case behaviour of programmable processors.

Beyond the quantum processor bound, we use additional input from quantum gravity: we assume that we cannot pass more than a black holes area worth of qubits into the black hole (a special case of the covariant entropy bound), and we use that the boundary CFT has a "short" description.[2] To reach the strongest version of our result, we will also make an assumption that a computation which is forbidden from happening inside a black hole also cannot be implemented inside of a smaller one.

Before proceeding, we note that another strategy to constrain bulk computation using the boundary description was suggested in [9], and similar ideas appear in [10,11]. That strategy involves noting that bulk computations are supported, in a sense that can be made precise, by boundary entanglement. The finite entanglement between distant boundary subregions can then be used to place constraints on the size of inputs for some bulk computations, and it has been further suggested that better understanding of entanglement requirements in non-local computation may lead to computational constraints.

## Summary of our thought experiment and result

The basic setting in which we constrain computation is shown in figure 1, where we consider a two sided black hole. A quantum system $A$ is recorded into bulk degrees of freedom $a$ and thrown into the black hole from the left asymptotic boundary, and a second system $P$ is recorded into bulk degrees of freedom $p$ and thrown in from the right. System $A$ initially holds a state $|\psi\rangle_A$, and $P$ holds a description of a unitary that needs to be performed, along with any computing device to be used to perform it. We will impose that the computer is built from a much smaller number of degrees of freedom than the black hole we are throwing it into, so that $n_p \ll S_{bh}$.[3] Otherwise, we can remain agnostic as to the design and functioning of this computer — it might exploit some exotic quantum gravitational effects in performing its computation. We aim to have the computer produce the state $\mathbf{U}|\psi\rangle_a$, which will be stored somewhere in the black hole. We assume that a global reconstruction of the $\mathcal{H}_a$ Hilbert space from the joint Hilbert space of both CFT's exists, and we require the reconstruction procedure is independent of the unitary to be performed.[4] Thus there is some isometry $\mathbf{R}$ that maps

---

[2]In particular we argue the CFT data can be specified in $O(\log(L_{AdS}/G_N)) = O(\log(c))$ bits.

[3]In the main text we will relax this to allow $n_p \leq S_{bh}$, which is enforced in the bulk by the covariant entropy bound. To do so and still find an interesting constraint, we will need to invoke the additional assumption that going to a smaller black hole never adds computational power. We consider the simpler setting where $n_p \ll S_{bh}$ in the introduction.

[4]As we will review, this is justified when $n_a + n_p \ll S_{bh}$.

$\mathcal{H}_A \otimes \mathcal{H}_{CFT} \rightarrow \mathcal{H}_A \otimes \mathcal{H}_E$, where $\mathcal{H}_A$ holds the state $\mathbf{U}|\psi\rangle_A$ if the bulk computation has succeeded.

To relate this setting to quantum information theory, consider the notion of a quantum programmable processor. An exact programmable processor is an isometry $\mathbf{T}$ which acts according to

$$\mathbf{T}_{AP \rightarrow AE}(|\psi\rangle_A |\phi_\mathbf{U}\rangle_P) = (\mathbf{U}_A |\psi\rangle_A)|\phi'_\mathbf{U}\rangle_E \, . \tag{1}$$

We will also consider approximate notions of a quantum processor. The $P$ Hilbert space holds a state $|\phi_\mathbf{U}\rangle$ which we call a program state, and which specifies a unitary $\mathbf{U}$ to be applied. We will consider non-universal programmable processors, which have program states for only some finite set of unitaries.

Returning to our black hole, we note that we can view the insertion of the relevant degree's of freedom, time evolution, and the recovery operation as the action of a quantum processor. This is because once the program state is prepared, the remaining operations used to carry out the computation — inserting these systems into the bulk, allowing the black hole to time evolve, then recovering the output system — are all independent of the program state, and can be viewed as a particular choice of isometry $\mathbf{T}$ that acts according to equation 1. We discuss the definition of $\mathbf{T}$ in more detail later on, but note here that it is fixed by the description of the CFT and of the initial state of the black hole.

Quantum processors are subject to constraints. Consider processors that implement a family of diagonal unitaries on $n_A$ qubits,

$$\mathcal{E} = \{\mathbf{U}^\varepsilon : \mathbf{U}^\varepsilon = \text{diag}(\varepsilon_1, \varepsilon_2, ..., \varepsilon_{2^{n_A}}), \varepsilon_i \in \pm 1\} \, . \tag{2}$$

For this family, one can show that an isometry $\mathbf{T}$ succeeds in implementing a randomly chosen unitary $\mathbf{U}^\varepsilon \in \mathcal{E}$ poorly whenever the number of qubits in the program state is sub-exponential in the number of data qubits. In particular, we will show that the probability $p(\mathbf{T}, \mathbf{U}^\varepsilon)$ of successfully applying the unitary[5] satisfies the bound

$$\mathbb{E}_\varepsilon \, p(\mathbf{T}, \mathbf{U}^\varepsilon) \leq C \frac{n_P}{2^{n_A}} \, , \tag{3}$$

where $n_P$ is the number of qubits in the program state, the average $\mathbb{E}_\varepsilon$ is over all values of $\varepsilon$, and $C$ is a constant.

Returning to the holographic setting, take

$$n_P \ll S_{bh} \, , \qquad \log C S_{bh} \leq n_A \ll S_{bh} \, . \tag{4}$$

The upper bound on $n_P$ is our imposition that we are considering a computer built of many fewer degrees of freedom than the black hole. We are free to choose $n_A$ as we like, and take $n_A \ll S_{bh}$ to ensure the inputs to the computation fit easily into the black hole. The lower bound on $n_A$ ensures $C n_P / 2^{n_A}$ will be small and our processor bound consequently non-trivial. Inside this regime, the bound 3 implies that some unitaries $\mathbf{U}^\varepsilon$ can be implemented in the bulk only with probability less than 1. By itself this is no surprise: to specify an arbitrary $\mathbf{U}^\varepsilon$ requires $2^{n_A}$ bits (the signs $\varepsilon_i$), so for some $\mathbf{U}^\varepsilon$ the program state of $n_P \ll S_{bh} \leq 2^{n_A}/C$ qubits is too few qubits to specify the unitary, preventing the bulk computer from applying it.

More surprising is that there are also unitaries with short descriptions that cannot be implemented in the bulk. To construct one, notice that the $\mathbf{U}^\varepsilon$ inherit an ordering from the strings $\varepsilon$. Choosing some threshold $\delta < 1$, we have from the bound 3 that some unitaries cannot be completed with probability higher than $\delta$. We define $\mathbf{U}^{\bar{\varepsilon}}$ as the first unitary which the processor $\mathbf{T}$ defined by our setting can't complete with probability more than $\delta$. In the main text we argue the CFT and the initial state can be efficiently described, using in particular $O(\log S_{bh})$ qubits, which means the description of these forbidden unitaries is small enough to be recorded

---

[5]We define this probability more precisely in the main text.

into $n_P$. Thus inside the black hole the computer holds a complete description of the unitary $\mathbf{U}^\varepsilon$ to be applied, but by construction the computer must fail to apply $\mathbf{U}^\varepsilon$, since otherwise the programmable processor $\mathbf{T}$ would succeed.

This construction shows that there are at least some computations which cannot be performed inside the black hole, despite there being no information theoretic reason they shouldn't be (i.e. the unitary is fully specified, and the inputs are available). Consequently, it is a computational restriction that forbids these unitaries from happening — we have shown that the bulk quantum gravity computer cannot implement arbitrary computations, and in particular cannot implement the explicit computation we constructed.

To better understand the workings of our bulk computer, it is interesting to ask how hard it is to implement the computations we've shown to be forbidden. In particular, what is their complexity, when considering for example a quantum circuit model of computation? We argue that in the regime 4, the computation that implements the needed unitary requires circuits with memory at least $CS_{bh}$ and depth at least $2^{S_{bh}}$. Assuming the physical computer has similar space and time requirements would suffice as a bulk explanation for why these computations are forbidden.

**Summary of notation**

We briefly recall some asymptotic notation used in computer science and employed here. We will use

$$f(x) = O(g(x)) \iff \lim_{x \to \infty} \frac{f(x)}{g(x)} < \infty,$$

$$f(x) = o(g(x)) \iff \lim_{x \to \infty} \frac{f(x)}{g(x)} = 0,$$

$$f(x) = \omega(g(x)) \iff \lim_{x \to \infty} \frac{f(x)}{g(x)} = \infty,$$

$$f(x) = \Theta(g(x)) \iff 0 < \lim_{x \to \infty} \frac{f(x)}{g(x)} < \infty.$$

In words, big $O$ means $f(x)$ grows not much faster than $g(x)$, little $o$ means $f(x)$ grows more slowly than $g(x)$, little $\omega$ means $f(x)$ grows faster than $g(x)$, and $\Theta$ means $g(x)$ and $f(x)$ grow at the same rate. Some other notation:

- We use capital Latin letters for quantum systems $A$, $B$, ..., except when they are bulk subsystems, in which case we use lower case Latin letters $a, b, ...$, etc.

- We use bold face capital Latin letters for unitaries and isometries, $\mathbf{T}$, $\mathbf{U}$, etc.

# 2 Programmable processors

In this section we define the notion of a programmable processor more carefully, then give a bound on a particular class of processors.

## 2.1 Universal and non-universal quantum processors

A classical computer functions according to the following basic structure. We input some data recorded in a string, call it $x$, and a program, call it $P$. Then the computer applies the program to the input data, producing output $P(x)$. When any program can be input to the computer in this way, we say the computer is universal.

In the quantum context the analogue is known as a universal processor. In this setting a program amounts to a specification of a unitary, and the input data is a quantum state. The overall action of a processor is given by an isometry $\mathbf{T}_{AP \to AE}$, which satisfies

$$\mathbf{T}_{AP \to AE}(|\psi\rangle_A \otimes |\phi_{\mathbf{U}}\rangle_P) = (\mathbf{U}_A |\psi\rangle_A) |\phi'_{\mathbf{U}}\rangle_E . \tag{5}$$

In [12], the notion of universal quantum processor was defined, and it was shown that for each distinct unitary (up to a phase) the processor can implement, an orthogonal program state is needed. Because there are an infinite number of distinct unitaries, no universal processor can exist in the exact setting.

Giving up on a universal quantum processor we can consider similar but weaker objects that might be possible to construct. One possibility is to consider approximate universal processors, allowing for some error tolerance in applying the unitary $\mathbf{U}$. Such approximate universal processors can be constructed [13], and it is known that any such construction needs the dimension of the program Hilbert space to scale exponentially with the dimension of the input Hilbert space [14]. Another route is to consider finite families of unitaries, and look for processors that apply only elements of this family, either exactly or approximately.

In this work, we will make use of results on this second notion of a quantum processor, which we now define more fully.

**Definition 1.** A quantum processor $\mathbf{T} : \mathcal{H}_A \otimes \mathcal{H}_P \to \mathcal{H}_A \otimes \mathcal{H}_E$ is said to implement the family of unitaries $\mathcal{U}$ if for each $\mathbf{U} \in \mathcal{U}$ there is a state $|\phi_{\mathbf{U}}\rangle \in \mathcal{H}_P$ such that

$$\mathrm{tr}_E \, \mathbf{T}(|\psi\rangle\langle\psi|_A \otimes |\phi_{\mathbf{U}}\rangle\langle\phi_{\mathbf{U}}|_P)\mathbf{T}^\dagger = \mathbf{U} |\psi\rangle\langle\psi| \mathbf{U}^\dagger , \tag{6}$$

holds for all $|\psi\rangle$. We also call such a construction a $\mathcal{U}$-processor.

To define a notion of an approximate $\mathcal{U}$-processor, one approach would be to require 6 holds approximately for all $\mathbf{U}$. Instead, we will define a quantity which captures how close to a $\mathcal{U}$-processor an isometry is in an averaged sense.

**Definition 2. (Processor testing scenario)** Consider an isometry $\mathbf{T} : \mathcal{H}_A \otimes \mathcal{H}_P \to \mathcal{H}_A \otimes \mathcal{H}_E$. The $\mathcal{U}$-processor testing scenario is as follows.

1. Choose $\mathbf{U}_A \in \mathcal{U}$ uniformly and at random.

2. Choose a state $|\phi_{\mathbf{U}}\rangle_P \in \mathcal{H}_P$. Apply $\mathbf{T}$ to $|\Psi\rangle_{\bar{A}A} \otimes |\phi_{\mathbf{U}}\rangle_P$, where $R$ is a reference system and $|\Psi\rangle_{\bar{A}A}$ is the maximally entangled state.

3. Measure the POVM $\{\mathbf{U}_A |\Psi\rangle\langle\Psi| \mathbf{U}_A^\dagger, \mathcal{I} - \mathbf{U}_A |\Psi\rangle\langle\Psi| \mathbf{U}_A^\dagger\}$.

The probability of passing this test is, using the optimal choice of program state, given by

$$p(\mathbf{T}, \mathcal{U}) \equiv \mathbb{E}_{\mathbf{U}_A \in \mathcal{U}} \sup_{|\phi_{\mathbf{U}}\rangle} \mathrm{tr}\big(\mathbf{U}_A |\Psi\rangle\langle\Psi| \mathbf{U}_A^\dagger \mathbf{T}(|\Psi\rangle\langle\Psi| \otimes |\phi_{\mathbf{U}}\rangle\langle\phi_{\mathbf{U}}|)\mathbf{T}^\dagger\big) . \tag{7}$$

The quantity $p(\mathbf{T}, \mathcal{U})$ gives our quantification of how close to a $\mathcal{U}$-processor $\mathbf{T}$ is.

## 2.2 Lower bounds on quantum processors

Below, we will show that $\mathcal{U}$-processors are constrained by the size of their program Hilbert spaces. We will be interested in processors implementing the family of unitaries

$$\mathcal{E} = \{\mathbf{U}^\varepsilon : \mathbf{U}^\varepsilon = \mathrm{diag}(\varepsilon_1, \varepsilon_2, ..., \varepsilon_{2^{2n}}), \varepsilon_i \in \pm 1\} . \tag{8}$$

This family is of particular interest because it can be related to the notion of type constants in the theory of Banach spaces, which will be the technical tool that eventually leads to our bound.

We now state the main claim of this section.

**Theorem 3.** *(Bound on $\mathcal{E}$-processors) Given an isometry $\mathbf{T} : \mathcal{H}_A \otimes \mathcal{H}_P \to \mathcal{H}_A \otimes \mathcal{H}_E$, we have*

$$p(\mathbf{T}, \mathcal{E}) \leq \frac{C \log d_P}{d_A}, \tag{9}$$

*where C is a constant.*

This will be the technical statement used in the next section, and the reader uninterested in the proof may proceed to there. In the rest of this section we explain some tools needed and then give the proof. Note that this result is similar to the bound given in [14], both in the techniques we will use to prove it and the statement. The only distinction is that in [14] they give a lower bound on the dimension of the program space in terms of a measure of the worst case performance of the processor. We can read the above as a lower bound on $d_P$ in terms of the performance of the processor on a particular state, the maximally entangled one, which can also be related to the average case performance of the processor.

The central mathematical structure we will exploit is the notion of a Banach space, and the theory of type constants. A Banach space $\mathcal{B}$ is a vector space equipped with a norm $||\cdot||_{\mathcal{B}}$, and which is complete under that norm. This can be compared to the more familiar notion of Hilbert space, which is a vector space with an inner product $\langle\cdot,\cdot\rangle$, and which is complete under the norm induced by that inner product $||x|| = \sqrt{\langle x, x \rangle}$. Notice that every Hilbert space is also a Banach space, but the reverse is not true.

Type constants are certain numerical values associated with a given Banach space $\mathcal{B}$ that characterize, in a sense we explain, how far from being a Hilbert space $\mathcal{B}$ is. In particular, if a norm is defined by an inner product, it carries with it additional structure beyond what is usually given by a norm. For example, in a Hilbert space we have

$$\frac{1}{2}(||x + y||^2 + ||x - y||^2) = ||x||^2 + ||y||^2. \tag{10}$$

How badly a Banach space can violate this equality then gives some notion of how far it is from being a Hilbert space. This motivates the following definition, which follows [15]. We will only exploit the type 2 constants, but give a more general definition for completeness.

**Definition 4.** Let $\mathcal{B}$ be a Banach space and let $1 \leq p \leq 2$. We say $\mathcal{B}$ is of type $p$ if there exists a positive constant $t$ such that for every natural number $n$ and every sequence $\{x_i\}_{i=1}^n$, $x_i \in \mathcal{B}$ we have

$$\left( \mathbb{E}_\varepsilon \left[ \left|\left| \sum_{i=1}^n \varepsilon_i x_i \right|\right|_{\mathcal{B}}^2 \right] \right)^{1/2} \leq t \left( \sum_{i=1}^n ||x_i||_{\mathcal{B}}^p \right)^{1/p}. \tag{11}$$

The infimum of the constants $t$ that satisfy this condition is the type $p$ constant of $\mathcal{B}$, which we denote $t_{\mathcal{B},p}$.

Note that in a Hilbert space $\mathcal{H}$, we always have $t_{\mathcal{H},2} = 1$.

It is also helpful to introduce the Banach space formed by linear operators acting on a Hilbert space. Given an operator $\mathcal{O} : \mathcal{H} \to \mathcal{H}'$ define the operator norm,

$$||\mathcal{O}||_\infty = \sup_{|\psi\rangle \in \text{Ball}(\mathcal{H})} ||\mathcal{O}|\psi\rangle||_{\mathcal{H}'}, \tag{12}$$

where $\text{Ball}(\mathcal{H})$ is the unit ball in Hilbert space $\mathcal{H}$. Then $\mathcal{L}(\mathcal{H}', \mathcal{H})$, the space of linear operators mapping $\mathcal{H}$ into $\mathcal{H}'$ which also have bounded operator norm, forms a Banach space. Considering the case of finite dimensional spaces, the type 2 constant of $\mathcal{L}(\mathcal{H}', \mathcal{H})$ can be bounded above according to [15, 16]

$$t_{\mathcal{L}(\mathcal{H}',\mathcal{H}),2} \leq C \sqrt{\min\{\log \dim \mathcal{H}, \log \dim \mathcal{H}'\}}. \tag{13}$$

With these ingredients, we are able to give the proof of theorem 3.

*Proof.* **(Of theorem 3)** We introduce the notation

$$|\Psi_\varepsilon\rangle_{AR} \equiv \mathbf{U}_A^\varepsilon |\Psi\rangle_{AR} \,,$$

and will denote the choice of program states by $|\phi_\varepsilon\rangle$. The success probability $p(\mathbf{T}, \mathcal{E})$ is expressed as

$$p(\mathbf{T},\mathcal{E}) = \mathbb{E}_\varepsilon \sup_{|\phi_\varepsilon\rangle} \mathrm{tr}\left[|\Psi_\varepsilon\rangle\langle\Psi_\varepsilon|(\mathbf{T}|\Psi\rangle\langle\Psi|\otimes|\phi_\varepsilon\rangle\langle\phi_\varepsilon|\mathbf{T}^\dagger)\right] = \mathbb{E}_\varepsilon \sup_{|\phi_\varepsilon\rangle} ||\langle\Psi_\varepsilon|\mathbf{T}(|\Psi\rangle\otimes|\phi_\varepsilon\rangle)||_E^2, \quad (14)$$

where $|||\psi\rangle_E||_E = \sqrt{\langle\psi|\psi\rangle}$ is the usual norm on the Hilbert space $\mathcal{H}_E$. Using that $|\Psi\rangle_{AR}$ is the maximally entangled state, and that

$$|\Psi_\varepsilon\rangle_{AR} = \frac{1}{\sqrt{d_A}}\sum_{i=1}^{d_A} \varepsilon_i |i\rangle_A |i\rangle_R \,, \quad (15)$$

we obtain

$$p(\mathbf{T},\mathcal{E}) = \frac{1}{d_A^2}\mathbb{E}_\varepsilon \sup_{|\phi_\varepsilon\rangle} \left|\left|\sum_{i=1}^{d_A} \varepsilon_i \langle i|_A \mathbf{T}(|i\rangle_A \otimes |\phi_\varepsilon\rangle_P)\right|\right|_E^2. \quad (16)$$

Define $\mathbf{T}_i \equiv \langle i|_A \mathbf{T}|i\rangle_A$, which is a linear map from $P$ to $E$. Then the above becomes

$$p(\mathbf{T},\mathcal{E}) = \frac{1}{d_A^2}\mathbb{E}_\varepsilon \sup_{|\phi_\varepsilon\rangle} \left|\left|\sum_{i=1}^{d_A} \varepsilon_i \mathbf{T}_i |\phi_\varepsilon\rangle_P\right|\right|_E^2 = \frac{1}{d_A^2}\mathbb{E}_\varepsilon \left|\left|\sum_{i=1}^{d_A} \varepsilon_i \mathbf{T}_i\right|\right|_\infty^2.$$

The last norm is on the Banach space of bounded linear operators from $\mathcal{H}_P$ to $\mathcal{H}_E$. Our choice of family of unitaries $\mathcal{E}$ has lead conveniently to the final expression here being the sum appearing in the definition of the type 2 constant. Using the result 13 for the upper bound on the type 2 constant of this Banach space, we obtain

$$p(\mathbf{T},\mathcal{E}) \leq C\frac{\log d_P}{d_A^2}\sum_{i=1}^{d_A}||\mathbf{T}_i||_\infty^2 \leq C\frac{\log d_P}{d_A}\,, \quad (17)$$

where we used that $||\mathbf{T}_i||_\infty \leq 1$ in the last inequality. This is exactly equation 9. $\qquad\square$

# 3 Forbidden computations for physical computers

In this section we relate bounds on programmable processors to computation in holographic spacetimes. Then, we comment on the interpretation of the resulting constraints from a bulk perspective. We begin however with a very brief review of some needed results in AdS/CFT related to reconstructing states in the bulk from the boundary.

## 3.1 The reconstruction wedge

A basic element in the understanding of AdS/CFT is the Ryu-Takayanagi formula, and its various generalizations and restatements. One form of the modern statement reads [17]

$$S(A) = \min_{\gamma_{ext}} \mathrm{ext}_{\gamma\in\mathrm{Hom}(A)}\left(\frac{\mathrm{area}(\gamma)}{4G_N} + S_{bulk}(E_\gamma)\right). \quad (18)$$

The area plus entropy term inside the brackets is often called the generalized entropy. The extremization is over surfaces $\gamma$ which are homologous to $A$, which means that there exists a codimension 1 surface $E_\gamma$ such that

$$\partial E_\gamma = A \cup \gamma \,. \quad (19)$$

The term $S_{bulk}(E_\gamma)$ counts the entropy inside the region $E_\gamma$. When there are multiple candidate extremal surfaces homologous to $A$, the final minimization picks out the one with least generalized entropy. The minimal extremal surface picked out by the optimization procedure in the RT formula will be labelled $\gamma_A$. This formula receives leading order corrections in some regimes, as understood in [18], but the form 18 will suffice for our application.[6]

Given a subregion of the boundary $A$, it is natural to ask if a subregion of the bulk is recorded into $A$. To make this question more precise, we should introduce a choice of bulk subspace, which we refer to as the code-space and label $\mathcal{H}_{code}$. The subspace $\mathcal{H}_{code}$ might for instance be specified by a particular choice of bulk geometry, along with some qubits distributed spatially across the bulk. Then, assume we are told the bulk degrees of freedom are in a state within $\mathcal{H}_{code}$, and we are given the degree's of freedom on subregion $A$. What portion of the bulk degrees of freedom can we recover?

Answering this question is related closely to the RT formula. In particular, the portion of the bulk we can recover if we know the bulk state in $\mathcal{H}_{code}$ is given by [19, 20]

$$E_A \equiv \bigcap_{\psi \in \mathcal{H}_{code}} E_{\gamma_A}. \tag{20}$$

That is, for each state in the code space we find where the RT surface $\gamma_A$ sits, and define the corresponding bulk subregion $E_{\gamma_A}$. Then, we define the intersection of all such surfaces, considering all states in the code-subspace. Note that in this procedure we should include mixed states of the code-space. The resulting region is the portion of the bulk degrees of freedom we can recover, if we know nothing about which state in the code-space the full bulk is in. This region is sometimes referred to as the *reconstruction wedge* of region $A$, defined relative to the code-space $\mathcal{H}_{code}$.

Given that it is possible to recover information inside the reconstruction wedge, we can also ask what explicit operation recovers the code space from the CFT degrees of freedom. Given a global map from the bulk subspace $\mathcal{H}_{code}$ to the boundary Hilbert space, it was understood in [21] how to construct such a recovery channel. Note that in this construction, a single choice of recovery channel works correctly for the entire code-space.

We will apply the notion of the reconstruction wedge with the region $A$ taken to be the entire boundary CFT. In this setting we might expect the reconstruction wedge is always the entire bulk, but if we choose too large of a code space it is possible for this to break down. In particular the minimal extremal surface appearing in equation 18 can appear that cuts out a portion of the bulk. While this incurs an area term with a cost like $L_{AdS}/G_N$ in the generalized entropy, if we take $\mathcal{H}_{code}$ large enough this can reduce the generalized entropy and be favoured. For this reason it will be necessary to keep our code spaces sufficiently small.

## 3.2 Holographic thought experiment with the game $G_{\mathcal{E}}$

Let's return to the setting of the thought experiment presented in the introduction. Our goal will be to construct a unitary acting on a small system that is forbidden from being completed in the black hole interior. It will also be important that the unitary have a short description: if even specifying the unitary requires an exponential number of bits, bringing this description into the region may itself induce a large backreaction and cause the experiment to fail.

To make the notion of an efficient description more precise, we recall the definition of Komolgorov complexity, also known as *descriptive complexity*. Intuitively, the descriptive complexity counts the minimal number of bits needed to describe a given string. Somewhat more formally, we make the following definition, which follows [22].

---

[6]Very roughly, in [18] it was understood that this formula breaks down when there are bulk states whose smooth max or min entropy differs at $O(1/G_N)$ from the von Neumann entropy, which won't occur here.

**Definition 5.** The **shortest description** of a string $x$ is the shortest string $\langle M, w \rangle$, where $M$ is a Turing machine and $w$ is an input string for that Turing machine, such that $M(w)$ outputs $x$. The **descriptive complexity** of $x$, which we denote $d(x)$ is the length of the shortest description.

Returning to holography, we consider two copies of a holographic CFT placed in the thermofield double state, so that the bulk description is a two sided black hole. We consider a one parameter family of black holes parameterized by their entropy $S_{bh}$.[7] We could realize this by for instance considering a family of CFT's parameterized by the central charge $c$, which is proportional to the black hole entropy. Our argument however is agnostic to how we realize this family, which we could also realize by adjusting the black hole temperature.

We are interested in putting constraints on what can be computed within an AdS space dual to a holographic CFT. Before proceeding, we should make some comments on what is meant by having performed a computation. Given some input system $\mathcal{H}_a$, we usually say that we have performed some computation $\mathbf{U}_a^\varepsilon$ (which will here be unitary) if the state on $\mathcal{H}_a$ transforms according to $|\psi\rangle_a \to \mathbf{U}_a^\varepsilon |\psi\rangle_a$. In quantum mechanics this is unambiguous, since the Hilbert space $\mathcal{H}_a$ is defined at all times. In field theory, we only have subregions of the spacetime, and a priori it's not clear what "the same" Hilbert space $\mathcal{H}_a$ at different times means. Unlike in quantum mechanics, we have different Hilbert spaces $\mathcal{H}_a$ and $\mathcal{H}_{a'}$ at early and late times, and some identification of bases in the two spaces. In practice we routinely identify persistent Hilbert spaces: for example we can track a given particle through spacetime, and call the Hilbert space describing its spin degree of freedom $\mathcal{H}_a$, but implicitly we have some basis information we are identifying across the early and late times.

In our context it will suffice to say that computation $\mathbf{U}^\varepsilon$ has been completed if we can identify in a "sufficiently simple" way a Hilbert space $\mathcal{H}_{a'}$ and identification of basis elements between $\mathcal{H}_a$ and $\mathcal{H}_{a'}$ such that the transformation $|\psi\rangle_a \to \mathbf{U}^\varepsilon |\psi\rangle_{a'}$ has been implemented. For us "sufficiently simple" will mean that the $\mathcal{H}_{a'}$ and the identification of bases can be specified using a number of bits small compared to other parameters in the problem. This agrees with the usual setting in quantum mechanics where $\mathcal{H}_{a'}$ is trivial to identify, and avoids some trivial ways of "performing" an arbitrary highly complexity computation, by for instance absorbing the computation into the basis identification. As an example, considering our particle moving through spacetime, we might identify the early and late time Hilbert spaces by specifying the background metric and parallel transporting a set of axes along the particle trajectory.

With this background on what we mean by a computation happening in a spacetime, let's proceed to understand the claimed constraints. We consider three agents, whom we call Alice, Bob, and the referee. The referee decides on some input size for the computation, call it $n_A = \log d_A$. We then play the following game.

**Definition 6. (Diagonal unitary game $\mathbf{G}_\varepsilon$)**

- Alice prepares a randomly chosen string $\varepsilon \in \{\pm 1\}^{d_A}$.

- Based on the value of $\varepsilon$, Alice prepares a state $|\phi_\varepsilon\rangle_P \in \mathcal{H}_P$ and acts on $\text{CFT}_R$ so as to record the state on $P$ into bulk degrees of freedom $p$, and throws this state into the black hole.

- The referee prepares the state $|\Psi\rangle_{\bar{A}A} = \frac{1}{\sqrt{d_A}} \sum_{i=1}^{d_A} |i\rangle_{\bar{A}} |i\rangle_A$ and gives the $A$ system to Bob. Bob acts on $\text{CFT}_L$ so as to record the state on $A$ into bulk degrees of freedom $a$, and throws this state into the black hole.

- Alice gives $\text{CFT}_R$ to the referee, Bob gives $\text{CFT}_L$ to the referee.

---

[7] In our asymptotic notation, e.g. $o(\cdot)$, the asymptotic parameter will always be $S_{bh}$.

- The referee applies a global reconstruction procedure on $\mathcal{H}_L \otimes \mathcal{H}_R$ to recover the state on the $a'$ system, which he records into $\mathcal{H}_A$. The Hilbert spaces $\mathcal{H}_a$ and $\mathcal{H}_{a'}$ should be identified as discussed above. The referee then measures the POVM $\{\mathbf{U}_A^\varepsilon |\Psi\rangle\langle\Psi|_{\overline{A}A}(\mathbf{U}_A^\varepsilon)^\dagger, \mathcal{I} - \mathbf{U}_A^\varepsilon |\Psi\rangle\langle\Psi|_{\overline{A}A}(\mathbf{U}_A^\varepsilon)^\dagger\}$.

If the referee obtains the measurement outcome $\mathbf{U}_A^\varepsilon |\Psi\rangle\langle\Psi|_{\overline{A}A}(\mathbf{U}_A^\varepsilon)^\dagger$, we declare Alice and Bob to have won the diagonal unitary game.

The steps in this procedure are summarized in figure 2.

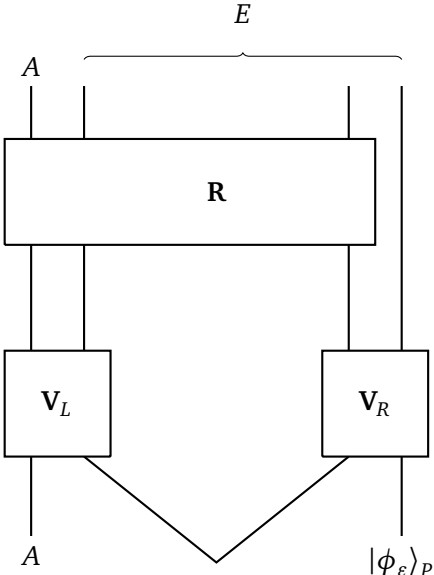

Figure 2: Circuit describing Alice and Bob's procedure to carry out the diagonal unitary game. Unitary $\mathbf{V}_L$ acts on $AL$, and corresponds in the holographic picture to recording the state on the $A$ system into bulk degrees of freedom $a$ sitting in the left asymptotic region. Unitary $\mathbf{V}_R$ acts on $RP$ and in the bulk picture corresponds to recording $P$ into degree's of freedom $p$ in the right asymptotic region. We allow the two CFT's to time evolve, which we absorb into $\mathbf{V}_L$ and $\mathbf{V}_R$, which in the bulk picture allows $a$ to interact with $p$. The isometry $\mathbf{R}$ extracts the $a$ system from the bulk and records it back into $A$. The state $|\phi_\varepsilon\rangle_P$ is prepared based on the string $\varepsilon$. The full circuit can be viewed as an isometry $\mathbf{T}_{AP \to AE}$.

In the reconstruction step, the referee applies a map $\mathbf{R}$ to the Hilbert space $\mathcal{H}_A \otimes \mathcal{H}_L \otimes \mathcal{H}_R$. We claim this map can be made independent of $\varepsilon$ and isometric. To understand why, recall from the last section that we can reconstruct $\mathcal{H}_{a'}$ in a state independent way if we take our code space to be the full Hilbert space of states that can depend on $\varepsilon$, since then the reconstruction procedure is independent of $\varepsilon$. Thus we should take $\mathcal{H}_{code}$ to include all of those states obtained by inserting any state in $\mathcal{H}_p$ and time evolving forward to the point where we do the reconstruction. If we would also like to reconstruct without holding the reference system $\overline{A}$, which we will need to apply our processor bound,[8] we should add in the $n_A$ qubits worth of states. Thus state independent reconstruction is possible when $n_A + n_P$ is much smaller than $S_{bh}$, so that the bulk entropy term never competes with the area of the black hole in finding the minimal extremal surface in equation 18. Concretely, it suffices to impose that

$$n_A + n_P = o(S_{bh}). \tag{21}$$

---

[8]This is because our processor bound is proven in the setting where $\mathbf{T}$ acts on $A$ and $P$ but not on the reference $\overline{A}$.

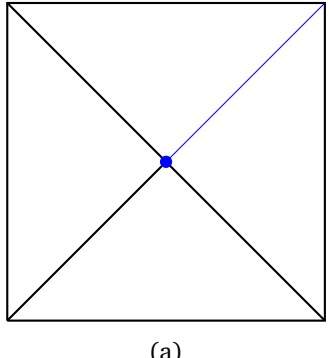
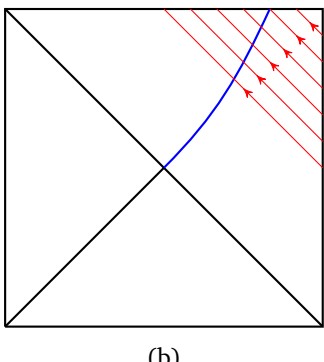

(a)                                                    (b)

Figure 3: a) We apply the CEB to the right going light sheet that begins on the bifurcation surface, call it $\Sigma$. Note that the information thrown in from the left will never cross $\Sigma$. b) Throwing in matter from the right deforms $\Sigma$. According to the CEB, $\Sigma$ will always bend inwards enough so that no more than area$(\Sigma)/4G_N$ qubits will cross through it. Consequently, information thrown in from the left will not encounter more than area$(\Sigma)/4G_N$ qubits thrown in from the right.

We will need to ensure we work in this regime.

The claim that **R** is isometric is easy to misunderstand in light of another set of ideas in AdS/CFT. Often it is useful to discuss the Hilbert space of an effective field theory that lives on the bulk geometry. In the context of black holes, this EFT Hilbert space is thought to map non-isometrically into the CFT Hilbert space [8]. Said another way, the EFT Hilbert space of black holes is too big, and some of its states do not have corresponding states in the fundamental (CFT) description. In our context we never introduce the larger bulk EFT Hilbert space. Instead, we begin with some CFT state dual to the two sided black hole, then act on the CFT to introduce the inputs to our computation. Thus our bulk state is necessarily a state in the fundamental description.

If indeed we can ensure **R** is state independent, we can notice that after the initial preparation of $\varepsilon$ all the steps in the protocol are independent of $\varepsilon$, and form an isometry. In fact, looking at the circuit diagram of figure 2 we see that the protocol is described by an isometry $\mathbf{T}_{AP \to AE}$ and a state preparation of $|\phi_\varepsilon\rangle_P$, which is then input to $\mathbf{T}_{AP \to AE}$. Thus the overall action is described by a map

$$\mathbf{T}_{AP \to AE}(|\Psi\rangle_{\overline{A}A} |\phi_\varepsilon\rangle_P) = \left|\Psi'\right\rangle_{\overline{A}A} \left|\phi'_\varepsilon\right\rangle_E . \tag{22}$$

This is exactly the action of a quantum programmable processor, so we have from theorem 3 that

$$p(\mathbf{T}, \mathcal{E}) \leq C \frac{n_P}{2^{n_A}} . \tag{23}$$

If we put appropriate constraints on $n_P$, $n_A$ this bound will lead to constraints on computation happening inside the black hole.

The value of $n_P$ we would like to have constrained physically, rather than as a choice we put in — $n_P$ controls the size of the computer, and we want to allow Alice and Bob to exploit the action of any physically allowed computer. A natural constraint on $n_P$ is given by the covariant entropy bound (CEB) [23–25]. We will apply the CEB to the bifurcation surface of the black hole, as shown in figure 3a. This limits the size of the computer that can be thrown into the hole according to

$$n_P \leq \frac{A_{bh}}{4G_N} = S_{bh} . \tag{24}$$

Notice that we can throw in arbitrarily large systems from the right and create a larger black hole, but at most $S_{bh}$ of these degrees of freedom can interact with the systems falling in from

the left. See figure 3b.

Unfortunately, at the upper limit of allowed values given by the CEB we violate 21, and lose our guarantee of state independent recovery. To continue our argument in light of this, we introduce an assumption, which is that if a computation is forbidden inside of a black hole with entropy $S'_{bh}$, then it is also forbidden inside of a black hole with entropy $S_{bh}$ with $S_{bh} = o(S'_{bh})$. That is, we will restore state independent recovery in the diagonal unitary game by allowing Alice and Bob an apparently more powerful resource, the geometry of a larger black hole, and assume this doesn't weaken their computational power.[9]

Now with the diagonal unitary game in the larger black hole in mind, consider the value of $n_A$. The value of $n_A$ is something we choose: we can decide to ask for a unitary on $n_A$ qubits to be applied inside the black hole, for whatever value of $n_A$. We will choose $n_A$ such that it is much smaller than $S_{bh}$, and so can be brought into the original black hole. Further, we will need to make $n_A$ large enough for equation 23 to be a meaningful constraint. Summarizing all the needed constraints, we consider running the diagonal unitary game inside of a black hole with entropy $S'_{bh}$, with $n_P, n_A$ satisfying

$$n_P \leq S_{bh} = o(S'_{bh}), \qquad \log(CS_{bh}) < n_A \leq S_{bh}. \tag{25}$$

In this regime, the constraint 21 is satisfied and the map $\mathbf{T}$ (which acts on the CFT state describing the larger black hole) is a state independent isometry. Consequently, the bound 23 applies, and using that $n_P \leq S_{bh} < 2^{n_A}/C$ we have that the average success probability of the diagonal unitary game will be below 1.

Now revisit the bound 23. Define the success probability of the processor $\mathbf{T}$ on value $\varepsilon$ as

$$p(\mathbf{T}, \mathcal{E}|\varepsilon) = \sup_{|\phi_\varepsilon\rangle} \text{tr}\left[ |\Psi_\varepsilon\rangle\langle\Psi_\varepsilon| \left(\mathbf{T}|\Psi\rangle\langle\Psi| \otimes |\phi_\varepsilon\rangle\langle\phi_\varepsilon| \mathbf{T}^\dagger\right) \right], \tag{26}$$

so that the processor bound 9 is expressed as

$$p(\mathbf{T}, \mathcal{E}) = \mathbb{E}_\varepsilon \, p(\mathbf{T}, \mathcal{E}|\varepsilon) \leq C \frac{n_P}{2^{n_A}}. \tag{27}$$

Setting some threshold probability $\delta$ with $Cn_P/2^{n_A} < \delta < 1$, we define the set

$$\mathcal{P}(\mathbf{T}, \mathcal{E}) = \{\varepsilon : p(\mathbf{T}, \mathcal{E}|\varepsilon) \leq \delta\}. \tag{28}$$

We refer to elements in this set as *forbidden unitaries*. From 27, this set will be of size at least

$$|\mathcal{P}(\mathbf{T}, \mathcal{E})| \geq 2^{2^{n_A}} \left(1 - \frac{Cn_P}{2^{n_A}\delta}\right), \tag{29}$$

which is doubly exponentially large in our parameter regime.

To understand the meaning of these forbidden unitaries, first notice that $n_A$ grows more slowly than $S_{bh}$. This means applying the needed unitaries is not restricted because the CEB is limiting the size of the systems acted on by our unitary. Looking at $n_P$ however, we can notice that since $n_P \leq S_{bh} < 2^{n_A}/C$, and $\varepsilon$ consists of $2^{n_A}$ bits, it is not possible to fit a complete description of an arbitrary $\varepsilon$ into $n_P$ qubits. If we can't even bring a specification of the unitary $\mathbf{U}^\varepsilon$ into the black hole, there's no surprise we can't implement it there — it's not possible to do so on information theoretic grounds. While this does explain why many unitaries are forbidden, we claim there are also some forbidden unitaries whose description can be compressed to fewer than $n_P$ bits. Consequently information theoretic constraints don't suffice to explain why those unitaries are forbidden.

We now define a unitary which both cannot be implemented in the bulk region, and has a short description.

---

[9]One way to argue for this is to consider that if the computation can be run inside the smaller black hole, we could take that black hole and throw it into the larger black hole, apparently running the same computation in the larger hole.

**Definition 7.** Define the unitary $\mathbf{U}^{\bar{\varepsilon}^0}$ to be the first element of $\mathcal{P}(\mathbf{T}, \mathcal{E})$, where the ordering is the one induced by interpreting the string $\varepsilon$ as a binary number.

Notice that from equation 29 the set $\mathcal{P}(\mathbf{T}, \mathcal{E})$ is non-empty and thus this unitary exists. Also observe that the above definition uniquely specifies this unitary.

We claim that $\mathbf{U}^{\bar{\varepsilon}^0}$ can be specified using $n_P$ bits, with $n_P$ inside of the regime 25. The definition above is an $\Theta(1)$ length string, plus the descriptive lengths of $\mathbf{T}_{AP \to AE}$ and $\mathcal{E}$. Let's consider the length of a description of each of these objects in turn.

- To describe $\mathcal{E}$, we need some $\Theta(1)$ description plus the value of $n_A$, which fixes the size of the unitaries in the set, which we can specify in $O(\log n_A)$ bits.

- To describe $\mathbf{T}_{AP \to AE}$ we need to specify $\mathbf{R}$ and the initial state in $\mathcal{H}_L \otimes \mathcal{H}_R$ appearing in figure 2.

    – To define the initial state of the two CFT's, we need to specify which CFT we are discussing, and the one parameter describing the black hole, for which we use the entropy $S'_{bh}$. Considering the description of the CFT, we assume there is a family of CFT's parameterized by the central charge $c$. Then to describe the CFT requires some $\Theta(1)$ data to specify which family we are considering, plus $\Theta(\log c) = \Theta(\log S'_{bh})$ bits to specify the member of that family. To specify $S'_{bh}$ requires at most $\log S'_{bh}$ bits.

    – Consider the map $\mathbf{R}$. This is fixed by defining the choice of CFT, the initial state of the CFT, and the choice of subspace $\mathcal{H}_{a'}$. The choice of CFT and initial state was already specified above. To specify the subspace $\mathcal{H}_{a'}$, recall that we defined having completed a computation to mean recording the output into a Hilbert space that can be described in a small number of bits. In the black hole context, we take this as meaning that we need far fewer than $S'_{bh}$ bits. We will allow in particular $\log S'_{bh}$ bits to specify the subspace.

The last point regarding the number of bits to specify $\mathcal{H}_{a'}$ is worth a few more comments. While we allow for $\log S'_{bh}$ bits, in the argument below anything smaller than $S'_{bh}$ bits will lead to forbidden bulk computations. Our specific choice of $\log S'_{bh}$ bits is motivated by considering the setting where, at the time of recovery, the bulk is described geometrically, and the output is recorded into some localized degree's of freedom. In this case we can specify the subspace using $O(\log S'_{bh})$ bits, since $S'_{bh}$ controls the size of the black hole and we would need to specify where in the black hole those bits are stored.

The full accounting then is that the descriptive length $d(\cdot)$ of $\mathbf{U}^{\bar{\varepsilon}^0}$ is

$$d(\mathbf{U}^{\bar{\varepsilon}^0}) = O(\log(S'_{bh}) + \log n_A) = O(\log S'_{bh}). \tag{30}$$

The second equality follows from our choice of parameter regime. From this equation, we see that we can describe $\bar{\varepsilon}^0$ using a state on $n_P$ bits whenever $\log S'_{bh} < S_{bh}$, which we can easily take while being consistent with $S_{bh} = o(S'_{bh})$. Notice also that we can define $\mathbf{U}^{\bar{\varepsilon}^m}$ as the $m$th element of $\mathcal{P}(\mathbf{T}, \mathcal{E})$, in which case we use $k = \log m$ additional bits. So long as we keep $k = o(S_{bh})$, this allows us to construct a family of unitaries of size $2^k$ which are similarly describable inside the black hole but forbidden from being implemented by the processor $\mathbf{T}$.

Let's summarize now our holographic thought experiment. On the right, Alice$_R$ prepares a randomly drawn string. Consider a case where she obtains a string describing a unitary in the set $\{\mathbf{U}_{\bar{\varepsilon}^m}\}_{m \leq 2^k}$. In this case, she can record a description of the unitary $\mathbf{U}^{\varepsilon}$ into no more than $S_{bh}$ bits. Doing so, and sending these bits into the black hole with larger entropy $S'_{bh}$, a complete description of $\mathbf{U}^{\bar{\varepsilon}^0}$ is inside the black hole. However, by construction these unitaries cannot be

completed with probability more than $\delta$ in our thought experiment. Thus performing these unitaries inside the black hole must be forbidden in the black hole of entropy $S'_{bh}$, and hence by our assumption forbidden inside the smaller black hole of entropy $S_{bh}$. In that setting, $n_P$ (the size of the computer) may be taken to be as large as the black hole entropy, $n_A$ (the size of the inputs) is still much smaller than the black hole, and the description of the forbidden unitaries is much smaller than $S_{bh}$, so can easily be brought into the black hole. Thus, the computation is forbidden from happening inside the smaller black hole using any physically allowed computer, even while the information needed to implement it is stored there — these forbidden computations must then be computationally forbidden. Further, there are at least $2^k$ such unitaries, with $k = o(S_{bh})$.

## 3.3 Bulk interpretation of forbidden unitaries

It is generally expected that the widely studied models of computation — classical Turing machines or quantum circuits — capture the power of physical computers. To make the connection between models of computation and physical computers, many authors have looked to gravitational constraints. This is because within quantum mechanics it does not seem possible to find a fundamental unit of time, or fundamental constraint on the memory held in a physical region.

As one example, Lloyd [26] offered a plausible gravity argument that, considering a circuit model of computation, the number of gates that can be performed in a given time is limited by the available energy. He then argues the available energy should be bounded above by the energy of a black hole, putting an apparent speed limit on computation. However, working with a Hamiltonian description of the computation one can evade this bound [2], doing arbitrarily complex operations arbitrarily quickly, and at arbitrarily low energy. While the needed Hamiltonians are likely unphysical, this construction shows that it remains unclear how to obtain a precise bound on computation from a direct gravity perspective.

Our construction of forbidden unitaries gives a very preliminary step towards connecting physical computers and models of computation: it at least shows that some computations cannot happen in certain finite spacetime regions. A natural question is how high of complexity our forbidden computations are, and if this high complexity offers some plausible physical reason from a bulk perspective why these unitaries should be forbidden.

We can make a few comments about the complexity of our forbidden computations. The needed computation is to, given the compressed description of $\overline{\varepsilon}^0$ and input system $A$, apply $\mathbf{U}^{\overline{\varepsilon}^0}$. One route to doing this is to first decompress $\overline{\varepsilon}^0$, then apply $\mathbf{U}^{\overline{\varepsilon}^0}$ based on the value of the uncompressed string. To decompress $\overline{\varepsilon}^0$ from its compressed description, we need to find the first value $\varepsilon$ where the function $p(\mathbf{T}, \mathcal{E}|\varepsilon)$ is smaller than $\delta$. A naive classical algorithm then to decompress the description of $\overline{\varepsilon}^0$ is the following.

$\varepsilon' = 0$
While $\varepsilon' \leq 2^{2^{n_A}}$
    If $p(\mathbf{T}, \mathcal{E}|\varepsilon') \leq \delta$,
        Return $\varepsilon'$
    Else
        $\varepsilon' = \varepsilon' + 1$

Assuming computing $p(\mathbf{T}, \mathcal{E}|\varepsilon')$ takes $O(1)$ steps (it is likely longer) this runs in $O(2^{2^{n_A}})$ steps. From 25 we see that this gives a number of steps in this algorithm of $2^{CS_{bh}}$. Further, notice that the memory needed to run this algorithm is at least the memory needed to store $\varepsilon'$, which is length $2^{n_A}$, so can be made as small as $CS_{bh}$ bits. In appendix A, we give a heuristic argument

that it is not possible to significantly improve on the memory usage and number of steps used in this algorithm, even using a quantum circuit model of computation.

The 'central dogma' of black hole physics states that black holes can be described as quantum mechanical systems with dimension $2^{S_{bh}}$. If we assume this, and assume a quantum circuit model captures the power of the bulk computer, this provides one plausible explanation for why these unitaries are forbidden in the bulk: the best quantum algorithm seems to require memory $CS_{bh} > S_{bh}$, so can't run inside the black hole.

We can also discuss the relationship between the number of computational steps needed to perform our unitary and the time available inside the black hole. Recall that we considered running our diagonal unitary game in the larger black hole of entropy $S'_{bh}$, where we first showed the computation was forbidden, assuming

$$n_P \leq S_{bh} = o(S'_{bh}). \tag{31}$$

Setting the above constraint amounts to a constraint on the choice of computing device thrown into the black hole, imposing that it is sufficiently small compared to the black hole entropy. Why does this constrained computer fail to implement the given computation in the larger black hole? Notice that the memory usage of the naive algorithm above is $2^{n_A} = CS_{bh} = o(S'_{bh})$, which is now much smaller than the black hole entropy. The number of computational steps of the naive algorithm now presents the most plausible computational restriction: the number of steps is $2^{2^{n_A}}$ which is much larger than $S'_{bh}$ if $\omega(\log S'_{bh}) = n_A$, which we are indeed guaranteed by our parameter regime 25. If we suppose a computational step takes some finite time, and that the naive algorithm above cannot be significantly improved in run time, this suffices as a bulk explanation for why our (restricted) computer cannot perform the needed computation. Because this seems to be the needed explanation in the context of the larger black hole, we might take this as evidence that the run time is also the relevant constraint in the (unconstrained) computer in the smaller black hole, although as noted above in that setting the memory is also larger than is available, again assuming a circuit model.

In fact, it is interesting to push this restriction on the size of the computer as far as possible and understand the number of computational steps needed in the resulting problem. Suppose we take $n_P = \log S'_{bh}$. This is the smallest we can take it while still allowing a description of $\mathbf{T}$ to be fit into $n_P$ bits. Then, we can have $n_A = \log(C \log S'_{bh})$ and still get a non-trivial bound from our processor bound. This leads to unitaries that are forbidden from happening inside of the black hole using a computer built from $n_P$ qubits. The memory then needed to run our naive algorithm is $\log S'_{bh}$, while the run-time is $S'_{bh}$. Thus the run-time of this small computer still seems to explain its inability to perform the computation inside the black hole.[10]

If we are willing to place constraints on the size of the computer by hand, there is no longer any need to consider the black hole setting, gave a natural surface on which to invoke the CEB. In the next section we consider restrictions on small computers in more general settings.

## 4 Forbidden computations for small computers

Given our construction in section 3 of constrained computations, we should ask to what extent our argument can be generalized away from the black hole setting, and away from AdS/CFT.

Towards making a more general statement, consider the following setting. We have a quantum mechanical system described by Hilbert space $\mathcal{H} = \mathcal{H}_A \otimes \mathcal{H}_P \otimes \mathcal{H}_E$ and evolving under Hamiltonian $\mathbf{H}$, where we refer to the $A$ system as the data Hilbert space, the $P$ system

---

[10]That our lower bound on the number of computational steps is exactly linear in the black hole entropy shouldn't be taken too seriously: since we are in an oracle model where evaluating $p(\mathbf{T}, \mathcal{E}|\varepsilon)$ takes one step, we are probably underestimating the run-time.

as the program space, and $E$ as the environment. Given a unitary $\mathbf{U}_A$, Alice prepares $\mathcal{H}_P$ in a state recording the unitary to be applied, or description of a program to apply it, along with any computing device prepared to apply it. She may use arbitrarily complex computations in preparing this state. Then, Bob prepares some state on the $A$ system. Further, the $E$ system is put in an arbitrary state $|\psi\rangle_E$ which we take to be initially pure, so that the environment is initially unentangled with the data and program spaces. The full Hilbert space is then allowed to evolve under time evolution given by the Hamiltonian $\mathbf{H}$. After some amount of time $t$, a measurement is made on the $A$ subsystem testing if $\mathbf{U}_A$ has been applied. This setting closely models the basic computational setting we find in the real world: we can prepare our computer which holds the program, insert the data, and then the computer runs — it evolves in this case under the Hamiltonian describing our universe.

In the black hole setting there is a natural bound on $n_P$, the number of qubits in the program space, which is imposed physically. In this scenario, we restrict $n_P$ arbitrarily — consequently, we are deriving here constraints on how fast *small* computers can perform computations, but not on all physically allowed computers. Also, note that in that setting the role of the environment Hilbert space $\mathcal{H}_E$ was played by the combined Hilbert spaces of the two CFT's.

Our processor bound 9 leads to a constraint on how quickly some unitaries can be performed in this scenario. In particular we have again that, after the system $P$ is put into the program state, the remaining action of the computer is described by an isometry independent of the unitary. In particular, the remaining action is just time evolution under $\mathbf{H}$. The description of $\mathbf{H}$, initial state of the environment $|\psi\rangle_E$, and amount of time we evolve for $t$ then defines a processor, which we label $\mathbf{T}$. Considering the family of unitaries 8, we can apply the processor bound 9, finding that

$$p(\mathbf{T}, \mathcal{E}) \leq C \frac{n_P}{2^{n_A}}\,. \tag{32}$$

Given an allowed program space of $n_P$ qubits, we choose the family of computations $\mathcal{E}$ such that $n_A$ is large enough, satisfying in particular

$$n_P \leq 2^{n_A}/C\,, \tag{33}$$

so that $p(\mathbf{T}, \mathcal{E})$ is less than 1. Given a value of $t$ and choice of Hamiltonian $\mathbf{H}$, we can then define a forbidden unitary in a way analogous to definition 7, which we do next.

Define the set of unitaries with low success probability

$$\mathcal{P}(\mathbf{T}, \mathcal{E}) = \{\varepsilon : p(\mathbf{T}, \mathcal{E}|\varepsilon) \leq \delta\}\,. \tag{34}$$

and then define a unitary which has a short description and is forbidden.

**Definition 8.** Let $\mathbf{U}_{\bar{\varepsilon}^0}$ be the first unitary in the set $\mathcal{P}(\mathbf{T}, \mathcal{E})$, where we order the set $\mathcal{E}$ by interpreting the strings $\varepsilon$ as binary numbers.

As before, we can also extend this to a family of unitaries.

How long is the description of $\mathbf{U}_{\bar{\varepsilon}^0}$? Importantly, it must be short enough to be written into $n_P$ qubits while maintaining $n_P \leq 2^{n_A}/C$. Notice that this definition consists of the $O(1)$ string given explicitly, plus a description of $\mathbf{T}$ and the parameter $n_A$ describing the set $\mathcal{E}$. Thus, if we have

$$d(\mathbf{H}) + d(|\psi\rangle_E) + \log t + \log n_A \leq n_P\,, \tag{35}$$

for $d(\mathbf{H})$ and $d(|\psi\rangle)$ the descriptive lengths of $\mathbf{H}$ and $|\psi\rangle$, there will exist forbidden unitaries which have descriptions fitting inside the program state, and hence must be computationally forbidden. We can always adjust our chosen value of $n_P$ to ensure this is the case.

The requirement above is essential to the physical consistency of our construction. One way this manifests is that we have

$$\log t \leq n_P \leq 2^{n_A}/C\,, \tag{36}$$

so that we cannot construct forbidden unitaries for arbitrarily small $n_A$ compared to $t$, which means the complexity of the computation cannot be made small compared to the time $t$. As an interesting case consider the setting where $\log t$ is much larger than the other parameters in the description of the isometry, in particular we allow a long enough time that

$$\log t \gg d(\mathbf{H}), \log n_A, d(|\psi\rangle_E). \tag{37}$$

Going to this setting, and using 36, we see that forbidden unitaries occur only for times shorter than $t \sim 2^{2^{n_A}}$. Recall that $2^{2^{n_A}}$ is exactly the scaling of the number of steps needed to decompress the forbidden $\varepsilon$. Thus our forbidden computations remain complex enough to ensure the number of steps it takes to implement them scales like the physical time needed to implement them on a computer.

Another comment is that we expect that for a given computation we can always find a $t$ large enough that our dynamical evolution implements the computation. Indeed our construction doesn't violate this, as it requires we first choose $t$, then can construct a unitary that cannot be implemented within time $t$. In particular we emphasize that for larger $t$ the value of $n_A$ must be chosen suitably large. A similar comment arises in comparing to the construction of Jordan [2]. Given a unitary, Jordan constructs a Hamiltonian that completes the unitary in an arbitrarily short time. In contrast, our ordering is different: we fix a Hamiltonian and a choice of time $t$ and then show there are computations that cannot be run by this Hamiltonian within that time. Since we expect there is ultimately one Hamiltonian describing our universe, this reversed statement seems sufficient to find physically unrealizable computations.

## 5 Discussion

In this work we have constructed computations which cannot be implemented inside of a black hole with entropy $S_{bh}$, despite the inputs to these computations being small, and the description of the computation being easily fit inside the black hole. We've argued that these computations are high complexity, which may explain why they are forbidden. Regardless of the explanation for why these computations are forbidden, our construction unambiguously establishes that at least some computations are forbidden from being implemented inside the black hole.

Moving forward, it would be interesting to understand general properties of unitaries that restrict their bulk implementation. To do this, we have two alternative approaches by which we can proceed. As we've done here, we can exploit the view of bulk computation in terms of programmable processors. Alternatively, following [9, 10, 27, 28], we can relate bulk computation to non-local quantum computation.[11] So far, the constraints coming from non-local computation have been complimentary to the ones derived from programmable processors. Perhaps one of these techniques, or some synthesis of the two, will allow further progress in the understanding of the limits of computation in the presence of gravity.

Before making a few comments on the connections between this work and others, we summarize the basic conceptual tension underlying our construction. A universal computer can follow instructions and, given an unbounded number of steps, perform any computation. Taking an outside view, and assuming our system is quantum mechanical, any computer evolves under the time evolution of some fixed Hamiltonian. This time evolution can be viewed as the action of a programmable quantum processor. Programmable processors are limited in the computations they can perform, while universal computers are apparently unrestricted, setting

---

[11]In an upcoming work [29], we adapt that discussion to the setting of two sided black holes. Doing so removes some assumptions made in [9], allowing constraints on non-local computation to be applied more rigorously to constrain the bulk. Another perspective on removing this assumption was presented in [11].

up a tension between the two perspectives. The naive resolution is that the programmable processor is only limited when the program states are small, restricting us from specifying most computations, thereby explaining on information theoretic grounds why the universal computer fails. For universal processors with simple descriptions however the tension becomes sharper — the universal computer can be input a description of the processor, which allows efficient descriptions of programs the computer can't itself run. Now, the way out of the tension is a computational restriction on the universal computer.

Our construction is similar to the diagonalization technique as used in computer science, in that the universal computer is being fed a description of the dynamics which it is itself is governed by. A key new ingredient however is the universal processor bound, which ties our argument to a physical setting. In particular, the length of the description of the processor, which relates to physical parameters (e.g. the time or black hole entropy), constrains the $n_P$ and $n_A$ parameters which then enter the processor bound. In this way physical data is brought into the diagonalization argument.

We conclude with a few comments on related topics.

### What is special about black holes?

We discussed here constraints on both computers inside of black holes and in ordinary AdS. We can briefly comment on what is unique about the black hole case. First, the black hole gave a natural covariant definition of a bulk subregion, and a surface on which to apply the CEB. These features are convenient but not strictly necessary: we could define a bulk subregion in some other way, and can then apply the CEB again or place a constraint on the size of the computer by hand. More fundamentally, the black hole gives us a way to specify the setting in a simple way, in terms of just the parameter $S_{bh}$. This parameter then appears in the specification of the forbidden computation, and controls the complexity of the forbidden computation. In contrast, away from the black hole setting, we had to specify a parameter $T$ setting the time for which we allow our system to evolve. The complexity of the forbidden computation is then set in terms of $T$.

### Quantum extended Church Turing thesis

The quantum extended Church Turing thesis states that any physically realizable computer can be efficiently simulated by a quantum Turing machine. Recently, Susskind [7] proposes an interesting tension with this thesis and a thought experiment in the setting of a two sided black hole. He argues that an observer who jumps into the black hole can compute certain functions efficiently that an observer who instead holds the two CFT's cannot. We find this thought experiment suggestive that a notion of an observer is needed in the statement of the extended Church Turing thesis, and the statement should only apply when two observers may separate for a time and then meet again and compare the efficiency of their computations.

While broadly this work and ours are both interested in the computational abilities of computers in the presence of gravity, we should be careful to distinguish between the two settings. Note that we never compare observers outside and inside the black hole and ask about their relative ability to perform some computation. Instead, we ask only about the computational abilities of the observer inside the hole. The boundary perspective is exploited to relate bulk computation to quantum processors.

### Complexity of the AdS/CFT dictionary

Recently, there have been discussions around the complexity of the operations needed to recover bulk data from the boundary [3, 6]. We emphasize that our argument does not rely on

this map being low or high complexity. Instead, we only rely on this map being state independent within some appropriate, and small, subspace of states.

**Bulk computation as non-local computation**

Our results are interesting in light of a conjecture made in the context of non-local computation and its relationship to AdS/CFT. Non-local computation implements unitaries on two, separated, subsystems using an entangled resource state and a single round of communication. In [9], the authors state that at least one of the following must be true:

1. All computations can be performed with linear entanglement.

2. Gravity places constraints on bulk computation.

They also argue that not 1) implies 2). That work conjectured that 1) is false and consequently 2) is true. This work establishes that 2) is true in AdS/CFT, without resolving 1).

**Understanding of the black hole interior**

In [8], the authors discuss a puzzle in the physics of black holes. The central dogma of black hole physics states that a black hole can be described by a number of degrees of freedom given by its entropy. The description of the black hole using $S_{bh}$ degrees of freedom is referred to as the fundamental description. Additionally, we can describe the black hole within effective field theory, within some background set by the appropriate solution to Einsteins equations. In the effective description, and at late times, the black hole interior volume can be very large. Thus the number of low energy degrees of freedom in the effective description will exceed $S_{bh}$. A puzzle then is to understand how the effective description, with a large number of apparent degrees of freedom, is embedded into the fundamental description with fewer degrees of freedom. Necessarily many of the states in the effective description will not be realizable states of the black hole, since most states cannot map to a state in the fundamental description.

To understand this, the authors of [8] argue that it is the low complexity states in the black hole interior that are mapped to the fundamental description. They show that even while the effective black hole interior is exponentially larger than the fundamental description, a subspace in the effective description large enough to contain all the low complexity states can be mapped to states in the fundamental description, and this map can approximately preserve orthogonality.

Our results support this perspective, in that they suggest high complexity unitaries are restricted in the bulk. In particular, the variation on our thought experiment most relevant to this discussion involves taking the computer to consist of $n_P = o(S'_{bh})$ qubits and considering the diagonal unitary game in the larger black hole, with entropy $S'_{bh}$. Then, the computer state is a state in the effective description of the black hole. Our argument then shows there are high complexity states the computer cannot evolve dynamically into, in line with the proposal of [8]. Said differently, our results support the idea that boundary time evolution, which must take fundamental states into fundamental states, also preserves a low-complexity set of states in the bulk.

**An end to time**

Among the strangest properties of black holes is that time in the interior comes to an apparent end at the singularity, at least within the classical description of the black hole. Understanding how this can arise from a quantum mechanical theory, in which time does not end, seems to be a basic challenge in understanding how gravitational physics can emerge from quantum mechanics. Our results support the idea that the finite bulk time corresponds, in some sense

to be made precise, to limits on bulk complexity enforced by the boundary theory: the bulk geometrizes the limits on complexity enforced by the boundary by having an end to time at the singularity.[12]

## Acknowledgments

We thank Michelle Xu, Patrick Hayden, Shreya Vardhan, Jonathan Oppenheim, Chris Akers, Jinzhao Wang, Raghu Mahadjan, Arvin Shahbazi Moghaddam, Steve Shenker and Toby Cubitt for helpful discussions.

**Funding information** AM is supported by the Simons Foundation It from Qubit collaboration, a PDF fellowship provided by Canada's National Science and Engineering Research council, and by Q-FARM. AMK and DPG are supported by the European Union (Horizon 2020 ERC Consolidator grant agreement No. 648913), the Spanish Ministry of Science and Innovation ("Severo Ochoa Programme for Centres of Excellence in R&D" CEX2019-000904-S, and grant PID2020-113523GB-I00), Comunidad de Madrid (QUITEMAD-CM P2018/TCS-4342), and the CSIC Quantum Technologies Platform PTI-001.

## A  Optimality of the naive algorithm

Recall that the full computation we wish to perform inside of the black hole is to apply $\mathbf{U}^{\bar{\epsilon}^0}$, given $\mathcal{H}_A$ and the compressed description of $\bar{\epsilon}^0$ as input. One method to do this is to first compute the full description of $\bar{\epsilon}^0$, then use this to apply the unitary. We will focus on algorithms of this form. Notice that within algorithms of this form we will always need memory of at least $2^{n_A}$, since that is the number of bits needed to store $\bar{\epsilon}^0$. A lower bound on the number of computational steps needed to compute $\bar{\epsilon}^0$ from its compressed description is less clear immediately, but we argue for one here.

The naive classical algorithm discussed in section 3.3 for decompressing the description of the forbidden unitary $\mathbf{U}^{\bar{\epsilon}^0}$ is as follows. $\varepsilon' = 0$

While $\varepsilon' \leq 2^{2^{n_A}}$
    If $p(\mathbf{T}, \mathcal{E} | \varepsilon') \leq \delta$,
        Return $\varepsilon'$
    Else
        $\varepsilon' = \varepsilon' + 1$

This uses $\Omega(2^{2^{n_A}})$ steps, and memory $\Omega(2^{n_A})$. One would additionally need the steps and memory necessary to then apply $\mathbf{U}^{\bar{\epsilon}^0}$.

Can we improve on this naive number of computational steps needed? We've seen that the function $p(\mathbf{T}, \mathcal{E} | \varepsilon)$ has at least some structure: equation 29 bounds how many input values on which $p(\mathbf{T}, \mathcal{E} | \varepsilon)$ is less than $\delta$, and that in fact the fraction of values where this function is less than $\delta$ is nearly one, being $1 - \Theta(n_P / 2^{n_A})$. Let's assume for a moment that this function has no additional structure aside from this condition. Thus, $p(\mathbf{T}, \mathcal{E} | \varepsilon)$ is treated as an oracle, with the promise that some fraction of its inputs return a function value less than $\delta$.[13] We count

---

[12]We thank Steve Shenker for making a similar remark to us.

[13]Note that, by equation 26, $p(\mathbf{T}, \mathcal{E} | \varepsilon)$ amounts to calculating the largest eigenvalue of the map $\langle \Psi | \mathbf{T}^\dagger | \Psi_\varepsilon \rangle \langle \Psi_\varepsilon | \mathbf{T} | \Psi \rangle : \mathcal{H}_P \to \mathcal{H}_P$. Hence, given access to the matrix elements $\langle i |_A \langle j |_P \mathbf{T}^\dagger | k \rangle \langle \ell |_A \mathbf{T} | r \rangle_A | s \rangle_P$ one could

one call to this oracle as a single computational step. How complex then is it to find the first input such that $p(\mathbf{T}, \mathcal{E}|\varepsilon) \leq \delta$?

To study this, define the Boolean function

$$f(\varepsilon) = \begin{cases} 1, & \text{if } p(\mathbf{T}, \mathcal{E}|\varepsilon) < \delta, \\ 0, & \text{if } p(\mathbf{T}, \mathcal{E}|\varepsilon) \geq \delta. \end{cases} \tag{A.1}$$

Let the number of inputs where $p(\mathbf{T}, \mathcal{E}|\varepsilon) \geq \delta$ be $N_s$. This can be as large as

$$N_s = 2^{2^{n_A}} \frac{n_P}{2^{n_A}}, \tag{A.2}$$

while maintaining consistency with equation 29, so let's assume this equality holds and that this is the only structure present in $f$. That is, the set of Boolean functions given by equation A.1 is assumed to be the set of all Boolean functions with exactly $N_s$ satisfying assignments. Restrict the function $f(\varepsilon)$ to its first $N_s$ possible inputs, so that now it is a function on $\log N_s$ bits, and call this function $\hat{f}(\varepsilon)$. Notice that finding the first $\varepsilon$ where $p(\mathbf{T}, \mathcal{E}|\varepsilon) < \delta$ is at least as hard as finding a satisfying solution to $\hat{f}(\varepsilon)$. But now the set of functions $\hat{f}$ is precisely the set of all possible Boolean functions on $\log N_s$ bits. Note that, unlike $f(\varepsilon)$, the function $\hat{f}(\varepsilon)$ has no restrictions on the number of inputs where it is 1 — it could have any number of satisfying inputs, including zero. Thus finding the first $\varepsilon$ such that $p(\mathbf{T}, \mathcal{E}|\varepsilon) < \delta$ is at least as hard the unstructured search problem with the oracle defined by $\hat{f}(\varepsilon)$. Using a quantum computer one can do no better than making $\sqrt{N_s} = \Omega(2^{2^{n_A-1}})$ oracle calls [30].

We can also briefly comment further on the memory usage needed in this problem. As mentioned at the beginning of this section, any algorithm where we first compute $\overline{\varepsilon}^0$ then apply $\mathbf{U}^{\overline{\epsilon}^0}$ we need $2^{n_A}$ bits of memory. However, we can also consider strategies that use some algorithm for applying $\mathbf{U}^{\overline{\epsilon}^0}$ that computes each bit of $\overline{\varepsilon}^0$ as it needs them, and erases each computed bit after it is used. Typically, finding such memory efficient algorithms comes at a cost to the number of steps, since many bits may need to be re-computed several times. Notice that in our setting to improve on the $2^{n_A}$ memory cost one would actually also have to do better in computational steps: any algorithm using $2^{2^{n_A}}$ steps will need $2^{n_A}$ bits of memory, since otherwise the algorithm will revisit one of its previous configurations and the computational will fall into an infinite loop. If we believe we cannot improve on the number of steps used in the naive algorithm then, we also cannot improve on the memory.

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
