# Peer review of "Constraints on physical computers in holographic spacetimes"

_SciPost Physics, doi:SciPost Phys. 16, 024 (2024)_

## Round 1 · Referee Report · Anonymous (Referee 1) · 2023-11-10

Strengths

1- Extremely clear 2-Studies a very interesting, relatively under-explored question 3-Invents an intriguing, original setup 4-Uses sophisticated, accurate calculations.

Weaknesses

1- In my view does not sufficiently motivate what's special about a black hole in their setup

Report

This journal's acceptance criteria are clearly met and I recommend for publication, after the authors consider the concern I make below. This is a clear, original, and highly interesting work.

My concern is the following: As presented, black holes do not seem special. However, I think there is a point you can emphasize more that will fix this problem.

Let me explain in more detail. When we say n_p, n_a < S_BH by the covariant entropy bound (CEB), we really mean S_BH of the final black hole, i.e. the black hole you end up with after it has eaten the n_p + n_a qubits. After all, there is nothing stopping us from throwing n >> S_BH qubits into a black hole of entropy S_BH in order to make a bigger one!

I claim this is important because it seems to show that these bounds are not special to black holes. Let me replace the black hole with simply m qubits floating in the middle of AdS (perhaps as a cloud of dust particles). Now we add our n_p + n_a qubits, ending up with n = m + n_p + n_a total qubits floating in the middle of AdS. I claim that the black hole bound n_p, n_a < S_BH is the same as the bound n_p, n_a < n here. It would follow that you have shown that there are unitaries that cannot be implemented on these n qubits in the same sense that you have shown that they cannot be implemented inside a black hole. In other words, there is nothing special about S_BH; what is special is n_p, n_a, and the restriction we have placed on them by hand.

That said, I think there is something special about black holes that you mention but could emphasize more. With black holes, unlike the cloud of qubits, you actually can satisfy n_p + n_a > S_bh by letting the black hole partially evaporate after throwing in the qubits. Then even with that black hole and its radiation in hand, it would be impossible to reconstruct them in a state-independent way and so the procedure must fail for that reason.

With this, you can make a pincering argument: if n_p, n_a < S_bh, these unitaries can’t be implemented by your current argument. If instead n_p + n_a > S_bh, then it can’t be implemented by this reconstruction-failure argument.

Then it seems clearer that S_BH is playing a distinguished role, and that black holes are different than a cloud of qubits.

Requested changes

1- Discuss the concern I raised above. What really makes black holes special in their argument? Is it important that black holes are special, in a way different from a cloud of qubits?

2- In equation (1.2), the A looks slightly out of place.

3- In Definition 2 step 1, the A subscript on U_A seems like it should be removed to be consistent with the rest of the steps.

4- I believe equation (2.6) needs a factor of 2 on the right hand side. And then below (2.7) you would want to say t_{H,2} = sqrt{2}, not 1.

5- Equation (2.13), probably want p(T, E) instead of just p(T).

6- Last sentence of the first full paragraph of page 10, “degree’s” should be “degrees”.

7- The sentence after footnote 7 is missing a word.

8- The sentence starting the paragraph at the bottom of page 12, a word is missing in “…is easy to misunderstood in light…”

9- Equation (3.12) seems to have an extra 2, and should be 2^{2^{n_a}} instead of 2^{2^{2n_a}}

  • validity: high
  • significance: high
  • originality: top
  • clarity: top
  • formatting: perfect
  • grammar: perfect

Author:  Alex May  on 2023-12-06  [id 4175]

(in reply to Report 1 on 2023-11-10)
Category:
answer to question

We would like to thank the referee for the careful and insightful comments.

We have addressed the more minor points raised by the referee, i.e. points 2-9 in their listing.

Regarding the more significant point about the CEB, and why it limits the number of qubits that the computer can be built from, we clarify this now around equation 3.7 and have added figure 3, which the referee may like to refer to. Roughly, the point is that while it is true we can add much more than S_{bh} (the orignal black hole entropy) qubits from the right into the bulk, and create a larger black hole, this doesn't let you interact those qubits with the matter falling in from the left. To interact with the left infallers, you need to cross the lightsheet defined by the bifurfaction surface, and the CEB limits the matter crossing that sheet to S_{bh}. Figure 3b illustrated gravitationally how this can be enforced. That this indeed is what happens at least in a simple example is shown in Bousso's "A covariant entropy conjecture" (1999) in section 6.

The wider point though was about what is special about black holes. We've added a brief comment on this in the discussion. One point is that the black hole gives a natural and covariant definition of a bulk subregion, and a natural surface on which to apply the CEB, though this isn't really essential (there would be various other ways to define the bulk subregion covariantly). The deeper point is that the black hole setting can be specified in a simple way, by just giving S_{bh} (and some details about which theory we are talking about, but that's O(1) bits), and the unitaries we construct that cannot be implemented inside the hole are then of a complexity related to the parameter S_{bh}. This should be contrasted with the non-black hole setting, where we had to specify a time parameter T, and the constructed forbidden unitaries then have complexity related to T.

I sort of speculative comment / perspective (which maybe explains why we've focussed on the black hole case) is that I find it much more interesting that there is a unitary with some complexity controlled by S_{bh} that is forbidden from the boundary perspective, since this seems to hint at the existence of the singularity, which is plausibly the bulk mechanism limiting the bulk computation. On the other hand the existence of a forbidden unitary with complexity that is some large value compared to T, the time we artificially limited the computer to run, seems like a less deep conclusion.

---

## Round 1 · Referee Report · Anonymous (Referee 2) · 2023-11-16

Strengths

  1. In a field with many proposed constraints on computation which are interesting but speculative, this paper gives a sharp rigorous constraint.

  2. I think the paper is mostly well-written, with important subtleties spelled out (such as ruling out unitaries which are allowed on information theoretic grounds).

  3. Although the technical details are somewhat complicated, I think the underlying construction has a simplicity and elegance to it which is appealing.

Weaknesses

  1. I found a few places in the discussion to be a little hard to understand, for example, the discussion around equation 3.7 constraining the value of nP.

  2. Similarly, I would have found a diagram or a little more detail to be helpful in section 3.3.

Report

I think the criteria are met because the paper "details a groundbreaking theoretical discovery" and "opens a new pathway". The paper reports a clearly argued and interesting result which, at least to me, suggests many new directions and connections. I strongly support publication.

Requested changes

See weakness comments.

---

## Round 2 · Referee Report · Anonymous (Referee 1) · 2023-12-14

Report

The authors have addressed my concerns in a satisfactory way. I continue to find it puzzling exactly how to interpret these results, but the authors have provided a good discussion of this point.

---

## Editorial Decision

published